# Traumatic Brain Injury Triggers Neurodegeneration in a Mildly Symptomatic MELAS Patient: Implications on the Detrimental Role of Damaged Mitochondria in Determining Head Trauma Sequalae in the General Population

**DOI:** 10.3390/metabo13010046

**Published:** 2022-12-28

**Authors:** Simona Zanotti, Daniele Velardo, Monica Sciacco

**Affiliations:** Neuromuscular and Rare Disease Unit, Fondazione IRCCS Ca’ Granda Ospedale Maggiore Policlinico, 20122 Milano, Italy

**Keywords:** TBI, MELAS, mitochondria, oxidative stress, antioxidants

## Abstract

Mitochondrial encephalomyopathy, lactic acidosis, and stroke-like episodes (MELAS) syndrome is a maternally inherited genetic mitochondrial disease with a typical onset in the first two decades of life and a major involvement of central nervous system (CNS). We present the case of a man affected with an oligosymptomatic, genetically determined MELAS syndrome, whose clinical picture dramatically and irreversibly worsened following a mild head injury. We hypothesize that the CNS metabolic stress induced by the brain injury activated an irreversible cascade of events leading to progressive neurodegeneration because damaged mitochondria were unable to restore the balance between energy requirements and availability.

## 1. Introduction

Mitochondrial encephalopathy, lactic acidosis, and Stroke-like episodes (MELAS) syndrome is a mitochondrial disorder which mainly affects the central nervous system and the skeletal muscle tissue, therefore resulting in a typical encephalomyopathy phenotype. It is a maternally inherited genetic disorder caused by mutations in mitochondrial DNA (mtDNA). A single base pair mutation, m.3243A > G, is found in 80% patients, and a second common mutation, m.3271T > C, is found in 10% [1,2].

Disease onset is usually in childhood or adolescence with non-pathognomonic symptoms such as seizures, headache, and or recurrent vomiting. The clinical diagnosis of MELAS, however, is based on more specific features, such as stroke-like episodes, usually occurring before the age of 40, encephalopathy with seizures and/or dementia, and evidence of lactic acidosis. The skeletal muscle biopsy shows ragged red fibres/RRFs [3,4].

Stroke-like neuroradiological lesions correlate with focal neurological symptoms including hemiparesis, hemianopia, or aphasia. Residual deficits from stroke-like episodes gradually impair neurologic function. At brain MRI/CT scan, they appear as infarct-like cortical areas which, however, do not correspond to any known vascular territory. Occipital or parietal lobes are most frequently involved at the beginning, and subsequent involvement includes the cerebellum, basal ganglia, and thalamus, followed by brain atrophy [5,6].

## 2. Case Report

A 40-year-old man came to our observation for clinical follow-up after receiving a genetic diagnosis of MELAS syndrome (m.3243A > G mtDNA mutation) from another institute. He was affected with diabetes and hypertension and, in addition, he referred to muscle cramps and easy fatiguability. Serum CK levels were slightly elevated. He had never had stroke-like episodes, nor did his brain CT scan show any typical lesions. He had no cognitive impairment (normal MMSE), epilepsy, or any other symptoms linked to CNS dysfunction. Indeed, he was well aware of his condition and pleased by the fact that, as a male, he would not transmit the disease to his children.

At the age of 43 years, he had a minor accidental occipital trauma followed by brief loss of consciousness and apparent full recovery with no evidence of acute damage at plain brain CT scan. After a few hours, however, he slipped into a deep coma (initial GCS 4) which lasted for over two weeks. He was dismissed from the hospital after 24 days. Upon awakening from the coma, he had started to manifest behavioural abnormalities and personality changes, including delirium with prominent sexual content, which prompted the administration of neuroleptic drugs. A slowly progressive mental decline was observed over the years, though the delusional state was pharmacologically controlled. Almost ten years later, however, the psychiatric manifestations got worse, and a severe cognitive impairment was diagnosed (MMSE always below 10). At the age of 56 years, he developed seizures, a progressive delusional state, visual and auditory hallucinations, and stereotyped actions mainly during the night (i.e., incessantly switching the light on and off, singing), all of which were never completely controlled by medications. An electroencephalogram (EEG) scan showed bilateral slow and paroxysmal abnormalities mainly localized in the left fronto-temporal brain regions.

Over the years, the patient developed progressive dysphagia and respiratory distress, marked neurosensory hearing loss, osteoporosis, and faecal and urinary incontinence. Frequent falls and diffuse tremors were reported. At the age of 61 years, MMSE could no longer be administered because he was unable to answer any questions. At the age of 62 years, he had a major left cerebrovascular ischemic stroke with clinical evidence of right hemiparesis, which was complicated three months later by septic shock due to a urinary infection. His situation became progressively worse and the patent died at the age of 65 years following a severe intestinal subocclusion, complicated, while at the hospital, by recurrent pneumonia.

## 3. Discussion

The clinical presentation of MELAS syndrome is very variable. The MELAS syndrome usually presents in adolescence or in early adult life. The case presented here refers to a patient with MELAS with very moderate clinical manifestations (diabetes and hypertension) without the occurrence of stroke-like episodes and no cognitive impairment at least until the age of 43. The patient’s clinical picture precipitated dramatically following an occipital trauma caused by an accidental fall.

We hypothesize that the head trauma triggered a sequence of events that led to the development of the MELAS disease in its full-blown form.

Given the elevated energetic request, optimal neuronal functions greatly rely on elevated ATP production by mitochondria and, consequently, mitochondrial dysfunction is at least partially responsible for the neuronal damage seen in neurodegenerative disorders including Parkinson’s disease (PD), Alzheimer’s disease (AD), Huntington’s disease (HD), and amyotrophic lateral sclerosis (ALS). The common denominator to this group of diseases is neuroinflammation, which triggers the release of proinflammatory mediators, namely cytokines and chemokines, able to activate microglial cells in the central nervous system (CNS). Activated microglia is a critical step in response to CNS injuries; however, when chronically activated, microglial cells can persistently produce toxic inflammatory mediators, namely reactive oxygen species (ROS), nitric oxide (NO), and various cytokines, which are able to inhibit mitochondrial function and therefore impair energy production, finally resulting in permanent neuronal injury [2,3,4,5,6,7].

A similar sequence of events has been proposed to explain the possible mechanisms involved in Traumatic Brain Injury (TBI) [8].

TBI can be defined as an alteration in brain functions caused by any direct blow to the head followed by impact of the brain against the skull. As in the case of neurodegenerative disorders, the impact is seen as a CNS insult that triggers an inflammatory response resulting in microglial activation and, when the activation is persistent, in a cascade of toxic responses ultimately leading to mitochondrial dysfunction and neuronal death. The mitochondrial damage results from an imbalance between fission, which is essential for the preservation of mitochondria numbers and location in case of energetic insults, and fusion, whose impairment, due to the inhibition of outer membrane fusion proteins mfn1 and mfn2, ultimately leads to the formation of undersized, low-functioning mitochondria [8]. Furthermore, it is important to note that mitophagy has been linked to aging. An impaired mitophagy, with a slower mitochondrial turnover by reduced mitochondrial biogenesis and inefficient mitochondrial degradation, was associated with age increase and is considered as a further aggravating factor, as reported also for several brain diseases such as cerebellar ataxia, Alzheimer’s disease and Parkinson’s disease [9].

Depending on the severity of the head injury, the dysfunction can be temporary or permanent, which is also the reason why TBI survivors can manifest emotional, cognitive, behavioural, or motor sequelae.

It is known that mitochondrial diseases manifest because damaged mitochondria fail to meet the energy requirements of tissues and organs, and that any situation of metabolic stress increases the energy demand, thus inducing additional damage, especially when a tissue with elevated energy requirements is involved [10].

Indeed, in 1999, Sharfstein and colleagues described the case of a still-undiagnosed female patient with MELAS syndrome whose clinical picture had been characterized only by frequent headaches and hearing loss until the age of 55 years, when she had been diagnosed with an ophthalmic herpes zoster infection and given oral prednisone for a few days. Five days after being treated, she had begun to experience progressive cognitive decline, which was initially misdiagnosed as herpes zoster encephalitis. She never recovered and developed severe dementia and acute psychotic events over the years, complicated by focal symptoms including hemiparesis, hemianopsia, and cortical blindness. The authors hypothesized that the febrile infection and the pharmacological treatment had triggered the MELAS manifestations in a previously compensated older patient [11].

Similarly, despite a MELAS diagnosis being made on the basis of non-invalidating endocrinologic and skeletal muscle symptoms, our patient had maintained a regular life until the age of 43 years when, he suffered what looked like a minor accidental head trauma with no neurologically documented brain injury. Recovery from the trauma coincided with the onset of severe and progressive behavioural and cognitive abnormalities, which were complicated over the following 20 years by seizures, a major stroke, and severe multiorgan dysfunction.

We do hypothesize that the CNS metabolic stress induced by the TBI activated the same irreversible cascade of events which is seen in primary neurodegenerative disorders due to the inability of damaged mitochondria to restore the balance between energy requirements and availability.

On a general basis, we suggest that any TBI, even a potentially mild one, occurring in a patient with mitochondrial dysfunction can trigger a degenerative pathway and cause severe, progressive, and irreversible damage.

This is a very important and significant argument to support and confirm the evidence that the prompt administration of antioxidants (i.e., ascorbic acid, carotenoids, CoQ10, N-acetylcysteine, glutathione, flavonoids, selenium, manganese) following TBI in the general population is able to inhibit ROS formation, thus preventing mitochondrial damage [12,13,14].

Moreover, it is known that, apart from genetically determined mitochondrial disorders, quite a few commonly used drugs and substances can have detrimental effects on mitochondria; indeed, they are not recommended in patients affected with proven mitochondrial diseases. The list includes, for example, antidiabetic, antiepileptic, antihypertensive, and antiarrhythmic drugs, as well as non-steroidal, anti-inflammatory drugs (NSAIDs), steroids, statins, chemotherapeutics, and immunomodulators. Ethanol and heavy metals, the latter often being seen as environmental contaminants, are also part of the list [15,16].

The effects of certain drugs and substances on mitochondria might explain why similarly severe TBI can have completely different sequelae on injured subjects. In this regard, an evaluation of the pharmacological records in TBI patients could be extremely useful in both retrospective and prospective studies of TBI patients to assert the neuroprotective role of antioxidants and to help establish TBI treatment guidelines.

## Data Availability

The data for this article are not publicly available to ensure patient anonymity. Requests to access the data should be directed to the corresponding author.

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
