# Peer review of "Traumatic Brain Injury Triggers Neurodegeneration in a Mildly Symptomatic MELAS Patient: Implications on the Detrimental Role of Damaged Mitochondria in Determining Head Trauma Sequalae in the General Population"

_metabolites, 2022, doi:10.3390/metabo13010046_

Round 1
Reviewer 1 Report
Dear Authors
I read with great interest your manuscript “Traumatic Brain Injury Triggers Neurodegeneration in a Mildly Symptomatic MELAS Patient: Further Evidence on the 3 Detrimental Role of Damaged Mitochondria in Determining 4 Head Trauma Sequalae in the General Population”
You reported a case of a 40 years old man with mildly symptomatic MELAS who fell into a 2-weeks coma after head trauma with subsequent progressive cognitive decline.
You said “minor accidental occipital trauma” with subsequent apparent fully recovery followed by coma. However, how patient fell into coma after minor head trauma is unexplained in your paper. Of note, a brain CT or MRI scans may persuade the readers about the entity of trauma and its possible related traumatic brain injury (TBI) with diffuse axonal injury or other lesions (e.g., cerebral oedema?).
It is common that patients with TBI shows behavioral abnormalities and personality changes after coma (delirium) which are well known risk factors for cognitive decline after coma in predisposed patients. In this scenario, hospitalization may further trigger behavioral changes and cognitive decline in a vulnerable patient such as this one. However, you said that severe cognitive decline was diagnosed 10 years later head trauma. How was the diagnosis done? Also, it would be useful for the readers providing the entity of the cognitive worsening by reporting MMSE before trauma, after coma as well as when “severe” cognitive decline was diagnosed.
I recommend reporting how long hospitalization lasted as well as any other potential factors which may contribute to delirium after coma and so trigger cognitive decline (e.g., laboratory testing etc).
In my opinion, without a clear neuroimaging data there are two major cause underlying cognitive decline in this predisposed patient which are hospitalization and coma.
Along the discussion you suppose that traumatic brain injury (TBI) is able to trigger cognitive decline. I agree. But TBI has not been reported for your patient.
The causality between “head trauma” without a clear TBI and clinical worsening in this patient cannot be demonstrated by simply reporting the clinical pipeline of the events. I recommend changing the title since this is not “further evidence”. This is not evidence.
Kind regards
Author Response
Dear,
We went through the observations and requirements made by the 1st reviewer and we answered accordingly:
Point 1. Of note, a brain CT or MRI scans may persuade the readers about the entity of trauma and its possible related traumatic brain injury (TBI) with diffuse axonal injury or other lesions (e.g., cerebral oedema?).
Response 1: Unfortunately, we do not have the possibility to insert any brain neuroimaging due to the time lapse between the traumatic event and the drafting of this Case Report (no major abnormalities reported, see below)
Point 2. How was the diagnosis done? Also, it would be useful for the readers providing the entity of the cognitive worsening by reporting MMSE before trauma, after coma as well as when “severe” cognitive decline was diagnosed
Response 2: Cognitive decline was diagnosed clinically and supported by MMSE evaluation. As requested, we Have now reported MMSE evaluation before and after the trauma as well as in the last period of his life, when cognitive decline had become extremely severe.
Section 2. Case Report, Line 56 and 69
Section 2. Case Report, Line 76 and 77
Point 3. I recommend reporting how long hospitalization lasted as well as any other potential factors
Response 3: We specified how long the patient had been hospitalized. Regarding laboratory tests, no abnormalities have ever been quoted in the documents given to the patient when he was discharged.
Section 2. Case Report, Line 62 and 63
In 2018, the Food and Drug Administration approved a blood test that detects two proteins, UCH-L1 and GFAP, which are released by the brain into the bloodstream when a mild concussion occurs. (How do healthcare providers diagnose traumatic brain injury (TBI)? NICHD - Eunice Kennedy Shriver National Institute of Child Health and Human Development, 2020). These tests were not yet available when the patient was hospitalized.
Point 4. It is common that patients with TBI shows behavioral abnormalities and personality changes after coma (delirium) which are well known risk factors for cognitive decline after coma in predisposed patients. In this scenario, hospitalization may further trigger behavioral changes and cognitive decline in a vulnerable patient such as this one. However, you said that severe cognitive decline was diagnosed 10 years later head trauma.
Response 4: Cognitive decline was diagnosed soon after he recovered from the trauma and got progressively worse over the years to the point that, after 10 years, the cognitive impairment became very severe.
This is specified in the text at section 2. Case Report at lines 65-66: “A slowly progressive mental decline was observed over the years…”
Point 5. The causality between “head trauma” without a clear TBI and clinical worsening in this patient cannot be demonstrated by simply reporting the clinical pipeline of the events. I recommend changing the title since this is not “further evidence”. This is not evidence.
Response 5: Traumatic brain injury is defined as an acquired insult to the brain from an external force that leads to temporary or permanent impairment of cognitive, physical, or psychosocial function (Gennarelli and Graham, 2005). Onset at the time of injury or within 24 hours corroborates the diagnosis. TBI severity is assessed using a combination of four factors: neuroimaging results (normal or abnormal), extent of altered or loss of consciousness, length of posttraumatic amnesia (up to 24 hours versus >24 hours), and Glasgow Coma Scale scores.
In our case, with the exception of neuroimaging results, which did not show major abnormalities, all other factors favor a diagnosis of severe TBI. Indeed, his mental situation dramatically and progressively worsened after the head trauma though no evidence of brain abnormalities had ever emerged before the traumatic event.
Genetic predisposition has been extensively studied both as a risk factor and as a negative prognostic index in patients with TBI and we believe that, in this case, the pre-existent, genetically determined mitochondrial damage in our patient had a major role in triggering the mental disturbance and in causing is irreversible progression.
Anyway, as requested by the reviewer, we changed the title by removing the words “further evidence” and replacing them with “Implications”
Please see the attachment: main text with the changes highlighted in green

Reviewer 2 Report
Given the elevated energetic request, optimal neuronal functions greatly rely on elevate ATP production by mitochondria and, consequently, mitochondrial dysfunction is at least partially responsible for the neuronal damage seen in neurodegenerative disorders including Parkinson’s Disease (PD), Alzheimer’s Disease (AD), Huntington’s Disease (HD) and amyotrophic lateral sclerosis (ALS). In this work, the authors hypothesized that the CNS metabolic stress induced by the TBI activated the same, irreversible cascade of events which is seen in primary neurodegenerative disorders due to the inability of damaged mitochondria to restore the balance between energy requirements and availability. The clinical presentation of MELAS syndrome is very variable. The MELAS syndrome usually presents in adolescence or in early adult life. The case presented here refers to a patient with MELAS with very moderate clinical manifestations (diabetes and hypertension) without occurrence of stroke‐like episodes and no cognitive impairment at least until the age of. The patient's clinical picture precipitated dramatically following an occipital trauma caused by an accidental fall. The authors hypothesized a dn show some evidence that the head trauma triggered a sequence of events that led to the development of the MELAS disease in its full‐blown form. I find this contribution of particular relevance for clinicians. However, I would emphasize the role of declining functionality of mitochondria in aging as a plausible aggarvating factor (see, doi: 10.3389/fnagi.2018.00032)
Author Response
Dear,
We went through the observation made by the 2nd reviewer and we answered accordingly:
Point 1. However, I would emphasize the role of declining functionality of mitochondria in aging as a plausible aggravating factor (see, doi:10.3389/fnagi.2018.00032)
Response 1: We thank the reviewer for the interesting insight. We have now inserted in the text the aspect of mitochondrial aging as a further possible factor of aggravation of the clinical picture.
Section 3. Discussion, lines 115-120
We have also added the appropriate referee (Wagner AP et al., 2018 as number [9]) in section 3. Discussion at line 120.
We have consequently changed the numbering of the bibliographic entries in section 3. Discussion and in References section
"Please see the attachment": main test with the changes highlighted in green

Round 2
Reviewer 1 Report
All the concerns have been addressed. You reported a case of cognitive decline triggered by TBI in patient with MELAS. This case might be useful for neurologists when inexplicable cognitive decline occur earlier in a predisposed patient.
Kind regards